# Surface Study of CuO Nanopetals by Advanced Nanocharacterization Techniques with Enhanced Optical and Catalytic Properties

**DOI:** 10.3390/nano10071298

**Published:** 2020-07-02

**Authors:** Muhammad Arif Khan, Nafarizal Nayan, Mohd Khairul Ahmad, Chin Fhong Soon

**Affiliations:** 1Microelectronics and Nanotechnology-Shamsuddin Research Centre (MiNT-SRC), Institute for Integrated Engineering, Universiti Tun Hussein Onn Malaysia (UTHM), Parit Raja, Batu Pahat Johor 86400, Malaysia; akhairul@uthm.edu.my (M.K.A.); soon@uthm.edu.my (S.C.F.); 2Department of Physics, Faculty of Basic and Applied Sciences, International Islamic University, Sector H-10, Islamabad 44000, Pakistan; shadiullahmarwat@gmail.com

**Keywords:** CuO nanopetals, surface study, advanced nanocharacterization, valence band & conduction band, catalytic activity, degradation, efficient charge separation

## Abstract

In the present work, a facile one-step hydrothermal synthesis of well-defined stabilized CuO nanopetals and its surface study by advanced nanocharacterization techniques for enhanced optical and catalytic properties has been investigated. Characterization by Transmission electron microscopy (TEM) analysis confirmed existence of high crystalline CuO nanopetals with average length and diameter of 1611.96 nm and 650.50 nm, respectively. The nanopetals are monodispersed with a large surface area, controlled morphology, and demonstrate the nanocrystalline nature with a monoclinic structure. The phase purity of the as-synthesized sample was confirmed by Raman spectroscopy and X-ray diffraction (XRD) patterns. A significantly wide absorption up to 800 nm and increased band gap were observed in CuO nanopetals. The valance band (VB) and conduction band (CB) positions at CuO surface are measured to be of +0.7 and −1.03 eV, respectively, using X-ray photoelectron spectroscopy (XPS), which would be very promising for efficient catalytic properties. Furthermore, the obtained CuO nanopetals in the presence of hydrogen peroxide ($H_{2}O_{2})$ achieved excellent catalytic activities for degradation of methylene blue (MB) under dark, with degradation rate > 99% after 90 min, which is significantly higher than reported in the literature. The enhanced catalytic activity was referred to the controlled morphology of monodispersed CuO nanopetals, co-operative role of $H_{2}O_{2}$ and energy band structure. This work contributes to a new approach for extensive application opportunities in environmental improvement.

## 1. Introduction

The development of the enhanced optical and catalytic properties in energy storage and environmental applications is mostly dependent on material fabrication [1,2,3]. Over the last few years, the results of metal oxide nanomaterials have shown remarkable efficiency due to its shape and size. Fabrication of excellence nanostructures of well-defined morphology and controllable size is an essential condition in order to prepare nanoscale devices or other numerous uses. Due to their nano-size, these materials have superior properties such as enhanced surface area and efficient charge transfer with better physical and chemical performance [4,5]. Furthermore, metal oxide nanomaterials have more active sites due to large surface area and have been considered as key materials, motivating researchers to divert their attention towards solving the problems in energy and environmental issues [6,7].

Copper oxide (CuO) has been considered to be one of the best semiconductor metal oxides as a result of its important chemical and physical properties such as large surface area, excellent solar light absorbance, and a narrow band gap (1.2 eV) [8,9,10,11]. CuO materials have a monoclinic structure and numerous attractive aspects, such as high stability, super thermal conductivity, photovoltaic properties, and antimicrobial activity. CuO material has the capability to transfer light in nature and has made a great contribution when used as a catalyst for dye removal in waste water treatment. Due to such unique properties, CuO nanoscale materials have attracted considerable attention for both fundamental scientific research and potential applications in technological fields. These fields includes solar cells, fuel cells, hydrogen storage devices, super capacitors, photo-catalysis, and catalytic applications [12,13,14,15]. These characteristics emphasize the need to understand the physical and chemical processes of CuO nanomaterials that take place at the surface and interfaces.

The CuO nanoscale materials having high surface area and uniform morphologies shall be helpful in enhancing capacity and conductivity of these materials. The control capability of the shapes and structures of copper oxide nanoscale materials has contributed significantly to improvement in the material’s fabrication [16,17,18]. Another important aspect of CuO nanostructure is that after fabrication, it is necessary to qualitatively and quantitatively assess nanomaterial properties so that the different morphology of nanostructured copper oxide would definitely result in production of special characteristics and more large-scale applications [19,20]. With the rapid advancement of nanotechnology, a different morphology of CuO nanostructures such as nanoplatelets, nanodendrites, nanocubes, nanoribbons, nanowires/nanorodes, leaf-like, and hollow structures, have recently been synthesized [21,22,23,24,25,26,27]. Among these highlighted nanostructures, well-defined stabilized monodispersed CuO nanopetal samples with controlled morphology will be the focal point for enhanced optical and catalytic applications.

Owing to environmental friendliness and higher electrochemical performance, CuO materials is considered one of the most significant environmental catalysts, capable of catalyzing hydrogen peroxide ($H_{2}O_{2}$) to generate very sensitive hydroxyl radicals (•OH). These hydroxyl radicals (•OH) can be used as an oxidizing agent to degrade a vast number of toxic substances (organic pollutants) without forming any harmful byproduct [28]. The degradation of organic pollutants using CuO nanostructures in the presence of $H_{2}O_{2}$ depends on many factors such as surface area and quantity of CuO nanocatalysts, concentration of $H_{2}O_{2}$, reaction temperature, and generation mode of of $H_{2}O_{2}$ [28,29,30]. Although CuO nanostructures with different shapes and architectures have been synthesized, the effect of the morphology of monodispersed stabilized CuO nanopetals by detailed advanced nanocharacterization and its effect as an environmental catalyst (CuO nanopetals) in the presence of $H_{2}O_{2}$ for the degradation of organic pollutants has been rarely explored [31]. To study and obtain well-defined stabilized monodispersed CuO nanostructures with significance enhancement in optical and potential catalytic activities, a better knowledge of the intrinsic nanomaterial properties is indispensable.

Herein, well-defined stabilized monodispersed CuO nanopetal samples with controlled morphology were synthesized by a simple hydrothermal technique and the results are described through a series of advanced nanocharacterization methods to investigate the enhancement of optical and catalytic properties. The hydrothermal method is the best technique to prepare the CuO petal-like nanostructure due to several advantages, such as low cost, simple setup, purity of obtained product due to the usage of water, repeatability and controllability, ability to reduce defects as compared to closed system having high temperature, and due to a superior approach for large-scale synthesis of mono-dispersed CuO nanopetals [18,27]. The structure, growth mechanism, optical and catalytic properties of the well-defined stabilized CuO nanopetals were studied in detail by advanced nanocharacterization techniques. High resolution transmission electron microscope (HR-TEM) equipped with selected area electron diffraction (SAED) were used for TEM and HR-TEM images and for the nucleation of CuO nanopetals along with their lattice spacing, plane, and direction. High-angle annular dark-field scanning transmission electron microscopy (HAADF-STEM) was used for atomic-resolution chemical mapping and energy-dispersive X-ray spectroscopy (EDS-STEM). The valance band (VB) and conduction band (CB) positions at CuO surface were measured using X-ray photoelectron spectroscopy (XPS). The phase purity of the as synthesized sample was confirmed by Raman and XRD patterns. The result of UV-VIS absorption spectra was observed to find the band gap of CuO nanopetals and suitable light for catalytic properties. A possible formation of CuO nanopetals and its reaction mechanism for the catalytic activities of copper oxide nanopetals in the occurrence of $H_{2}O_{2}\;$ for the degradation of methylene blue under dark is proposed. 

## 2. Materials and Methods 

### 2.1. Materials

Commercial, high-grade copper nitrate trihydrate $\;\left[\mathrm{Cu}\;\left({\mathrm{NO}}_{3}{)}_{2}\cdot 3H_{2}O\right)\right]$, 99.95%, hydrogen peroxide ($H_{2}O_{2}$, 30%), methylene blue (MB), acetone $\left(C_{3}H_{6}O\right)$, ethanol $\left(C_{2}H_{5}\mathrm{OH}\right)$, and Hexamethylenetetramine (HMTA,$\;C_{6}H_{12}N_{4}$ ≥ 99.0%) were obtained from Sigma–Aldrich and Merck KGaA Germany. Analytical grade chemicals, which need no further purification, were used throughout the experimental study. In this experiment, all chemical solutions were synthesized in pure water, which is obtained from water a purifying system (micro HIQ). Also, de-ionized distilled water is used for cleaning and nitrogen stream is used for drying the samples.

### 2.2. Synthesis of Copper Oxide Nanopetals

The synthetic experiment of CuO petal-like nanostructures were prepared through a simple hydrothermal method. For the growth of CuO NPs, a modified approach in a basic medium was used with controlled growth parameters [32]. The fabrication method is shown in Scheme 1, where copper oxide nanopetals are prepared by the hydrothermal method. Specifically, 5 mM copper nitrate $\left[\mathrm{Cu}\;\left({\mathrm{NO}}_{3}{)}_{2}\cdot 3H_{2}O\right)\right]$ powder was dispersed in 100 mL deionized water and then mixed with 1 mM solution (aqueous) of HMT (100 mL) under stirring for 60 min at room temperature. Subsequently, the suspension mixture was shifted within an autoclave (75 mL) and sealed. The sealed autoclave then was enclosed in an oven at 110 °C for 3 h. After cooling to room temperature, the solution was centrifuged and washed with DI water to reduce the pH to below 9. Finally, a product of black color was obtained after drying in a flow of nitrogen through a vacuum oven for 2 h at 60 °C. 

### 2.3. Characterization

The MIRA3 TESCAN FE-SEM well equipped with EDS operating at an accelerating voltage of 15 kV was used for the morphology and elemental analysis, respectively. X-ray diffraction (XRD) using Geol diffractometer with CuKα (0.154 nm) at 40 kV and 20 mA was used for the structural information of CuO nanopetals. JEOL JEM-ARM 200F transmission electron microscope was used to characterize the TEM and HR-TEM at an acceleration voltage of 200 KV. The growth and nucleation of CuO nanopetals along with their lattice spacing, plane, and direction are confirmed by HR-TEM (JEOL JEM-ARM 200F) operating at 200 kV. TEM is also used for the selected area electron diffraction (SEAD) and advanced STEM analysis. A HAADF-STEM image of a single CuO nanopetal confirmed the elemental composition and crystal information at atomic scale. The bonding behavior and phase purity of the as synthesized sample was studied by Raman Spectroscopy (HORIBA Scientific) at laser wavelength 525 nm characterization techniques at 365 nm laser source. X-ray photoelectron spectrometer (XPS) (Shimadzu Kratos Axis Ultra DLD) using current value 10 mA at 15 kV and carbon reference peak at 284.60 eV were used for the binding energy spectra and to study the valance band and core level spectra. UV-VIS spectrometers (Schimadzu 1800) were used for the analysis of optical properties. UV–VIS spectrophotometer equipment was used for the light absorption, transmittance and for the band gap calculations. Also, the absorption of the degraded MB solution was analyzed by UV-VIS spectrometers at room temperature.

### 2.4. Catalytic Testing

The catalytic activity of CuO petal-like nanostructures was determined by degrading MB in aqueous solution. For the catalytic testing measurement, primarily, the stability of the degradation dyes MB was tested with $H_{2}O_{2}$ only in dark, and their outcomes unmistakably identified that the degradation dyes have a more stable nature under dark. In a typical catalytic test, 20 mg of the as-grown CuO nanopetals were taken in powder form and mixed with 100 mL of an aqueous solution of MB (0.2 g/L) at room temperature under constant stirring. In addition, 20 mL of $H_{2}O_{2}$ (30 wt%) solution was added to the MB solution containing CuO nanopetals and the mixture was allowed to react for 60 min under stirring to reach equilibrium. Then at a regular time interval, 3.5 mL of reaction solution in quartz cuvette was removed and analyzed by UV–visible spectrophotometer. A standard calibration curve for methylene blue concentrations was achieved by calculating the peak intensity at max ʎ=664.64 nm.

## 3. Results and Discussion

### 3.1. Surface Morphology

FE-SEM characterization technique is used to examine the surface/interface morphology. The surface morphology of the prepared materials plays a dynamic role to increase the efficiency of catalytic properties. The evidence of dimension for the fabricated CuO nanopetals was analyzed with the help of FE-SEM. The FE-SEM images of the as-synthesized CuO nanopetals with different magnification are presented in Figure 1. FE-SEM images of CuO nanopetals as shown in Figure 1a,b demonstrates that large scale monodispersed CuO nanopetals can be produced via our synthetic approach. Figure 1c presents the high-resolution FE-SEM image, which shows that copper oxide nanopetals are monodispersed and have a well-defined morphology. The monodispersed and well-defined morphology means that CuO nanopetals have the lack of aggregation between the individual nanopetal and are perfectly controlled in size, shape, and internal structure. The length and diameters of the CuO nanopetals are varied from 1.5–1.7 µm and 600–700 nm, respectively. The high magnification FE-SEM image clearly shows that CuO nanopetals are well fabricated, highly stable, monodispersed, and have a large surface area with very sharp tips. Furthermore, the perfect size of CuO nanopetals were examined with HR-TEM measurements. The X-ray energy-dispersive spectroscopy (EDS) attached with field emission scanning electron microscope was used to find the corresponding energy-dispersive X-ray (EDX) spectrum and elemental analysis of CuO nanopetals. Only the presence of Cu and oxygen elements were confirmed. The EDX spectrum of CuO nanopetals is shown in Figure 1d and their atomic and weight percentage of Cu and O elements are tabulated in the inset in Figure 1d.

### 3.2. Crystallinity

The crystallinity of the as-synthesized CuO nanopetals was checked by X-ray diffraction and the result of the XRD pattern is shown in Figure 2. It can be clearly observed from the XRD diffraction pattern that all peaks belong to the pure monoclinic crystalline of CuO nanopetals, which is matched with the standard card number of ICSD 98-006-9757. The strong peaks of CuO nanopetals are observed at 2θ values of $35.6^{{}^{\circ}}$ and $\;38.8{\;}^{{}^{\circ}}$, which are referred to as the planes of (-111)-(002) and (111)-(200), respectively. The peak position of these planes are the nature of monoclinic CuO crystallites, which belong to the pure phase of CuO nanopetals. The standard XRD pattern of monoclinic phase CuO obtained from ICSD 98-006-9757 database has been compared with the XRD pattern of CuO nanopetals, as shown in the inset of Figure 2. It clearly shows the absence of characteristic impurity peaks in the XRD pattern of CuO nanopetals. Furthermore, no other additional product (impurity), such as ${\mathrm{Cu}}_{2}O$ or $\;\mathrm{Cu}{\left(\mathrm{OH}\right)}_{2}$, can be observed. The evidence obtained from XRD pattern indicated that our synthesis procedure is pure, controllable, and capable of reducing defects compared to a closed system.

### 3.3. X-rays Photoelectron Spectroscopy Analysis

XPS analysis is a powerful surface sensitive technique that has been used to confirm the chemical composition, purity, and oxidation state of CuO nanopetals. X-rays photoelectron spectroscopy measurements were performed using a scanning X-ray microprobe (Ulvac-PHI, INC) with monochromatic source Al Kα radiation at photon energy 1486.6 eV and current value 10 mA at 15 kV for the binding energy (BE) spectra. The C 1s (carbon 1s) peak at 284.60 eV was used as a reference for the calibration of all the binding energies. The survey or wide energy range scan (low resolution) analysis of XPS spectrum taken from a CuO nanopetals sample was measured with a pass energy of 160 eV and step interval (smallest energy division) of 1 eV for the entire sample surface. For a region core level or narrow energy range scan (high-resolution) analysis, XPS spectra were performed with a passing energy of 20 eV and step interval of 0.1 eV.

Figure 3a shows XPS wide scan spectra (survey spectrum) of CuO nanopetals, which have confirmed that the peaks are associated with the elements of Copper (Cu), Oxygen (O), and Carbon (C). Figure 3b–d show the high-resolution spectra (core XPS spectra) of Cu 2p, O1s, and C 1s, respectively. The XPS results can differentiate CuO from both ${\mathrm{Cu}}_{2}O$ or metallic copper by presence of the dominant shake-up peak as shown in Figure 3b. The core level or narrow energy range spectra of Cu 2p shows main shake-up peak at the higher binding BE side of the $\mathrm{Cu}\;2p_{3/2}$ and the increase in BE of the main peak, suggesting the existence of an unfilled Cu3d9 shell. The existence of an unfilled Cu3d9 shell further confirmed the presence of ${\mathrm{Cu}}^{2+}$ in the sample of the CuO [7]. Furthermore, in the core level spectra of Cu 2p, the peaks at 953.7 eV and 933.6 eV could be assigned to ${\mathrm{Cu}}^{2+}2p_{1/2}$ and $\;{\mathrm{Cu}}^{2+}2p_{3/2}$ of CuO, respectively. Figure 3c shows the de-convolution of high-resolution XPS spectra for O1s in CuO nanopetals. The Gaussian-Lorentzian fitting of asymmetric peak of O1s shows three components at 529.1, 530.78.5, and 532.21 eV. The peak at 529.1 eV, which is more intense, can be assigned to the binding energy for lattice oxygen $\;{\left(O_{L}\right)}^{2-}$ in CuO lattice and is in good agreement with the binding energy of $O^{2-}$ ion in the metal oxide sites (${\mathrm{Cu}}^{2+}-O^{2-})$ [33]. The second peak at 530.78 eV can be assigned to the binding energy for oxygen defects/vacancies $\;{\left(O_{V}\right)}^{2-}$ within the matrix of CuO. The last peak at 532.21 eV can be assigned to the binding energy for adsorbed residual carbon or other surface oxygen species, which easily react with the CuO nanopetal surface [34].

Figure 3d shows the high-resolution spectra of carbon (C 1s), which confirmed the referenced peak at 284.67 eV and another higher energy peak at 288.41 eV. Both peaks of C 1s spectrum known as adventitious carbon contamination, which is commonly used as a charge reference for XPS spectra on the surface of the sample. The C 1s referenced peak with binding energy 284.67 eV, assigned to adventitious carbon containing C–C bond, while C 1s peak with binding energy 288.41eV, assigned to adventitious carbon containing O–C=O bond. The interpretation of XPS Spectra of CuO nanopetals has also confirmed, that that there is no residual nitrogen from the starting materials. The results measured from the XPS spectra verify CuO nanopetal structure.

### 3.4. Raman Analysis

The room temperature Raman analysis of CuO nanopetals was performed by Raman spectroscopy technique within Raman shift from 200–800 cm^−1^, as shown in Figure 4. CuO has a monoclinic structure with space group symmetry of $C_{2h}^{6}$. There are 12 zone-center optical phonon modes having three acoustic modes $\left(A_{u}+2B_{u}\right)$, six infrared active modes $\left(3A_{u}+3B_{u}\right)$ and three active Raman active modes $\left(A_{g}+2B_{g}\right)$ as shown in Equation (1) [35].
(1)$$\Gamma_{RA}=4A_{u}+5B_{u}+A_{g}+2B_{g}$$

Figure 4 shows that CuO nanopetals consist of three Raman active optical phonons at 296, 346, and 631 cm^−1^. The broad peak in comparison with other peaks has a high intensity assigned to $A_{g}\;$ band at 296 cm^−1^, while the other two peaks in comparison with $A_{g}$ band are low intensity, assigned to $2B_{g}$ at 346 and 631 cm^−1^. Copper oxide (CuO) nanopetals having a single phase with a monoclinic structure was determined as a result of Raman analysis. No other ${\mathrm{Cu}}_{2}O$ modes or impurities are displayed, validating that CuO nanopetals have a single phase with a monoclinic structure. The symbolic intensities of the three peaks also evidenced that CuO nanopetals having monoclinic structure with a pure CuO phase show good agreement with the previous reports [36,37,38]. The preparation method of CuO nanopetals, its geometry and crystal structure also play an important role in determination of the Raman peaks position.

### 3.5. TEM Analysis

TEM analysis is one of the more advanced nanocharacterization techniques used to determine the perfect morphologies and crystal quality of the as-grown CuO nanopetals. The CuO nanopetal samples data obtained from TEM analysis were studied in detail for (i) low magnification images, (ii) high magnification images, (iii) diffraction pattern (SAED), and (iv) energy dispersive spectrum (EDS). The low magnification TEM images usually describe the particles size, diameter, length, distribution, and agglomeration of observed sample. Figure 5a shows the low magnification TEM image of CuO nanopetals on TEM grid. The TEM image demonstrates that large scale synthesized monodispersed CuO nanopetals with large surface area and well defined (uniform) morphology can be produced via our synthetic approach. The low magnification TEM result is fully consistent with the FE-SEM results. Figure 5b,c present the low magnification TEM images of individual (single) CuO nanopetals at magnification 500 nm and 200 nm, respectively. The length and diameter of the CuO nanopetals were measured using Gatan Microscopy Suite 3.0 (GSM 3.0), and are 1611.96 nm and 650.50 nm, respectively. The TEM images of single CuO nanopetal at magnification 100 nm and 50 nm shown in Figure 5d–f, indicate clear and smooth surface with sharp tips. Figure 5f shows the growth directions of single CuO nanopetals along [100] and [010] planes. The preferred accumulated growth direction (length) of CuO nanopetals is along [010] plane as compared to [100] plane (width), because [010] is the least substantial plane due to lowest density of copper atoms, while [100] plane is more stable compared to [010] plane due to the highest density of copper atom. The TEM images CuO nanopetals as shown in Figure 5b–d indicates well-defined structures. These structure of individual CuO nanopetals has shown that the core of the particle is darker than the borders part (bright area). The border part (bright area) is the main part of the CuO nanopetal, while the core part (dark domain) is the side part, which interpenetrating grow with the main part. It means that the TEM image presents a well-defined structure with accumulation of an interpenetrating sheet-like nanostructure. The high magnification TEM images generally describe the crystallinity, atomic size, thickness/layer distance, oxidation/contamination, and defects/boundaries of the observed sample. Figure 5g,h shows HR-TEM images of a single CuO nanopetal, clearly verifying that the as-synthesized CuO nanopetals were pure crystalline. The spacing between two neighboring fringes of CuO nanopetals was measured as ~ 0.27 nm using Joel Gatan Microscopy Suite 3.0 (GSM 3.0), which is comparable to the distance of the (110) plane of the monoclinic CuO as shown in Figure 5h. Furthermore, no flower-shaped or agglomeration of petal structures were identified during TEM samples analysis, which support the understanding of monodispersed and stabilized CuO nanopetals by our synthesis techniques. The selected area electron diffraction (SAED) pattern generally describes the nature of crystallinity and dominant crystalline/structure of the observed sample. Figure 5i is a SAED pattern of CuO nanopetals taken in the normal direction from the rectangular part as shown in Figure 5f, showing polycrystalline nature of CuO nanopetals. It is worth noting that each part of CuO nanopetals presents a single crystalline pattern with monoclinic phase of CuO, as shown in HR-TEM image of Figure 5h, while SAED pattern revealed that all parts of CuO nanopetals grown along its length direction [010] and width direction [100]. Also, it is observed that diffraction spots are stretched in radial direction, which support the idea that nanopetals consist of nanocrystals having a large density with a slightly different orientation. The TEM analysis also supports the Raman and XRD results.

### 3.6. Advanced STEM Analysis (HAADF-STEM)

HAADF-STEM is another powerful tool of TEM analysis for the characterization of nanostructures and imaging of various material interfaces, which gives the elemental composition and crystal information at atomic scale [39,40]. The scanning TEM works by concentrating an electron beam inside a very small size spot, scanned over the TEM sample. In STEM analysis of CuO nanopetals, using the high-angular annular dark field (HAADF) mode, one is capable to gain atomic-resolution featured images, known as bright field image as shown in Figure 6a,b. Figure 6c illustrates the single CuO nanopetal bright field image, and its corresponding EDX mapping in STEM of CuO nanopetals is sketched in Figure 6c. The chemical mapping of elemental composition confirmed the presence of net Cu and O elements with the same kind of distribution. The STEM-EDS pattern of single CuO nanopetals as shown in Figure 6d also support FE-SEM analysis.

### 3.7. Optical Properties

The energy structure and optical properties of CuO petal-like nanostructures are of interest because of their catalyst and semiconductor characteristics. The optical properties of CuO petal-like nanostructures were investigated using a UV-visible spectrometer for its light absorption, transmittance, and band gap calculation. Figure 7a shows UV–visible absorption spectrum of as prepared copper oxide nanopetals. The CuO nanopetals indicate an active and large absorption of light in the visible region. Furthermore, the band-gap energy of CuO nanopetals samples can be found according to Equation (2) [41].
(2)$$\alpha h\nu =A{\left(h\nu -E_{g}\right)}^{n/2}$$
where α = absorption coefficient, *h* = Planck constant, *ν* = light frequency $E_{g}\;$= band gap, *A* = constant and *n* = electron transition between conduction and valance band, *n* = 1 (allowed direct transition), *n* = 4 (allowed indirect transition), *n* = 3 (forbidden direct transition) and *n* = 6 (forbidden indirect transition). The plot of ${\left(\alpha h\nu \right)}^{2}$ versus energy  (hν) for determining $E_{g}$ (energy band gap) has been shown in Figure 7b. The band gap energy of the copper oxide petal-like nanostructure was found by deducing the linear part of $\;{\left(\alpha h\nu \right)}^{2}$ versus energy  (hν). The calculated band gap energy of CuO nanopetals is about ≈1.73 eV, which is different from the bulk CuO (1.24 eV). Yang et al. [30] reported the band gaps of the ellipsoid-like, plate-like, boat-like, and flower-like CuO nanostructures were determined to be 1.371, 1.447, 1.429, 1.425 eV, respectively [30]. The measured band gap value of CuO nanopetals is greater than the band gap values with closely related structures and materials available in the literature. This change further supported an active and large absorption due to the nanometer range of synthesized material. This result suggests that the band gap changes in CuO nanopetal material could display higher catalytic activity. Figure 7c shows transmittance spectra of CuO nanopetals, which show a weak transmittance in the entire visible range. The weak transmittance in the entire visible range also support the higher absorption in that region.

In addition, valance band (VB) position of CuO petal-like nanostructure was further investigated using XPS wide spectra as shown in Figure 7d. The valence band position (VB) was determined to be +0.70 eV for the CuO nanopetals sample. Considering the band gap energy of CuO nanopetals shown in Figure 7b and VB position shown in Figure 7d, the conducting band (CB) positions of CuO nanopetal sample was calculated from the following formula in Equation (3) [42].
(3)$$E_{CB}=E_{VB}-E_{g}$$

The conducting band position (CB) of −1.03 eV was obtained for CuO nanopetals sample, which would be very promising for energy storage and efficient catalytic properties.

### 3.8. Catalytic Properties

The catalytic activities of CuO petal-like nanostructure for the degradation of methylene blue (MB) was checked in the presence of $H_{2}O_{2}$ at room temperature without using any specific light source (external source). The absorption spectra of the degradation of MB were studied using UV-VIS spectroscopy instrument. The amounts of methylene blue,${\;H}_{2}O_{2}$ and copper oxide nanopetals were used fixed as described in Section 2.4 of catalytic testing. Figure 8 shows the chemical structure of methylene blue, with characteristic absorption peak at around 664.64 nm, which was thoroughly observed at various time intervals to check out the degradation process of MB dye. Figure 9a,b shows typical time dependent UV-visible absorption spectra of methylene blue solution in the presence of $H_{2}O_{2}$ and $H_{2}O_{2}+\mathrm{CuO}$, respectively. During the course of degradation, the MB solution was first measured only in the presence of $H_{2}O_{2}$ at room temperature as shown in Figure 9a. The MB solution was degraded to only ~ 7% through 90 min. This proves that the MB solution in the presence of $H_{2}O_{2}$, without addition of any CuO nanopetals was degraded, showing very small change within a short time. Figure 9b shows typical time dependent UV-visible absorption spectra of MB solution in the presence of $H_{2}O_{2}$ and CuO nanopetals. Throughout degradation, the color of the MB solution underwent fading and intensity of absorption decreased regularly, suggesting a sharp decrease of MB solution. Finally, the characteristic absorption peak of MB solution became too broad and weakened through 90 min, implying almost complete degradation of MB. The concentration ratio plot of the MB solution in the presence of $H_{2}O_{2}$ versus specific time intervals and MB solution in the presence of $H_{2}O_{2}+\mathrm{CuO}$ versus specific time intervals are shown in Figure 9c. The efficiency, known as effectiveness of the degradation (percentage of decolorization) of CuO nanopetals, were determined by the following formula in Equation (4) [43].
(4)$$ƞ=\left(1-\frac{S_{t}}{S_{0}}\right)\times 100\%$$
where ƞ = % D = percentage of dye degradation, $S_{t}$ = absorbance (concentrations) values of MB dye at the *t*^*t*ℎ minutes, and $S_{0}$ = absorbance (concentrations) values of MB dye at the 0^*t*ℎ minutes. The MB was almost 99% degraded, when CuO petals were applied in the aqueous solution of MB + $H_{2}O_{2}$ as shown in Figure 9d. The CuO nanopetals in the presence of $H_{2}O_{2}$ has enhanced the degradation efficiency and maximum degradation efficiency of MB to > 99% within 90 min, which presents a significant enhanced catalytic effect of copper oxide nanopetals. The enhanced catalytic activity is due to well-defined large surface area of stable CuO nanopetals, efficient charge separation due to the co-operative role of hydrogen peroxide, and band structure. The particular copper oxide nanopetals have deep importance, and can be used as a good catalyst at room temperature without using any external source of light.

The reaction mechanism of the fast degradation of methylene blue with the support of hydrogen peroxide and CuO nanopetals without using any external source of irradiation with photon proceeds according to the following [45].

Step 1.
$$\left[\mathrm{Cu}\left(\mathrm{II}\right)\right]+H_{2}O_{2}⇌\left[\mathrm{Cu}\left(\mathrm{II}\right)\right]\ldots .H_{2}O_{2}\to \left[\mathrm{Cu}\left(I\right)\right]+•\mathrm{OOH}+H^{+}$$
$$\left[\mathrm{Cu}\left(I\right)\right]+H_{2}O_{2}⇌\left[\mathrm{Cu}\left(I\right)\right]\ldots .H_{2}O_{2}\to \left[\mathrm{Cu}\left(\mathrm{II}\right)\right]+•\mathrm{OH}+{\mathrm{HO}}^{-}$$

Step 2.
$$•\mathrm{OOH}+H_{2}O_{2}\to H_{2}O+•\mathrm{OH}+O_{2}$$
$$•\mathrm{OH}+H_{2}O_{2}\to H_{2}O+•\mathrm{OOH}$$

After adsorbing methylene blue molecule and $H_{2}O_{2}$ on the surface of the CuO nanopetals, $H_{2}O_{2}\;$ reacts with the complex surface of the petals (nanostructure) [Cu(II)], which produce as a result free radical •OOH and species  [Cu(I)]. Further reaction with $H_{2}O_{2}$, oxidized back to [Cu(II)] in conjunction with radical  •OH as shown in Step 1. Step 2 represents that the free radicals may once again be adsorbed on $H_{2}O_{2}$ and produce free radicals of HO, HOO, or •O^2−^. The free radicals produced as a result of Step 1 and 2 are responsible for very high oxidizing ability to interact with MB dye and therefore greatly enhance the oxidative degradation rate of methylene blue dye.

Furthermore, degradation of methylene blue with the support of hydrogen peroxide and CuO nanopetals follow the pseudo-first-order kinetics [44], since there is a linear relationship between ln(S_t_/S_o_) and t (time) as shown in Equation (5).
(5)$$ln\left(\frac{S_{t}}{S_{0}}\right)=-kt$$
where *k* = pseudo first order constant, *t* = reaction time, $S_{t}$ = absorbance (concentrations) values of MB dye at the *t*^*t*ℎ minutes, and $S_{0}$ = absorbance (concentrations) values of MB dye at the 0^*t*ℎ minute. Figure 9e shows the linear plot, ln (St/So) versus t (time) for the degradation of methylene blue (MB) using copper oxide nanopetals. The linear plot in Figure 9e had excellent linear correlations,${\;R}^{2}=0.998$ implying that the degradation reaction follows Equation (4). The value of k (0.044$\;\min^{-1}$) was found from the slope of the linear line, which further support higher catalytic efficiency of copper oxide nanopetals.

The comparative study of catalytic properties of CuO nanopetals in the presence of $H_{2}O_{2}$ for the degradation of MB dye with some earlier reported works are given in Table 1, which clearly point out the importance of the present results due to the following. First, the current results of CuO nanopetals in the presence of $H_{2}O_{2}\;$ has shown an efficient catalytic response (degradation efficiency) compared with the other reported works shown in Table 1. Second, a significantly wide absorption up to 800 nm and increased band gap were observed in CuO nanopetals. Third, the valance band (VB) and conduction band (CB) positions at the CuO surface are measured for the first time, which was found to be +0.7 and −1.03 eV, respectively, using X-ray photoelectron spectroscopy (XPS), which would be very promising for efficient catalytic properties. 

The enhanced catalytic activity was referred to the controlled morphology of monodispersed CuO nanopetals, co-operative role of $H_{2}O_{2}$, and energy band structure. The CuO nanopetals are the lack of aggregation between the individual nanopetals and are perfectly controlled in size, shape, and internal structure. CuO nanopetal catalysts in the presence of hydrogen peroxides form reduction and oxidation pairs to degrade the MB dye. Free radical and species of HO, HOO, or •O^2−^ are produced due to the catalytic decomposition of hydrogen peroxide. These free radicals and species are the dominant oxidation agent, which are responsible for efficient degradation of MB dye.

In addition to measuring the stability of the catalytic activity of copper oxide nanopetals, the catalytic test for the degradation efficiency of MB dye in the presence of hydrogen peroxide $\left(H_{2}O_{2}\right)$ and CuO nanopetals was repeated five times. The efficiency results of CuO nanopetals in the presence of $H_{2}O_{2}$ for the degradation of MB dye for the process repeated five times are shown in Figure 9f, which show no significant variation in the MB degradation rate, confirming that the CuO nanopetals have good stability. Furthermore, the CuO nanopetals as a catalyst will be utilized in the environmental waste water treatment.

## 4. Conclusions

Novel copper oxide petal-like nanostructures were well prepared using the hydrothermal technique. Various advanced nanocharacterization tools were used to describe dominant morphological, structural, and elemental composition of copper oxide nanopetals. The advanced TEM analysis including HRTEM, SAED, and HAADF-STEM characterization explored that the CuO nanopetals grew a substantial amount, offered controlled morphology, were monodispersed with large surface area, and show the nanocrystalline nature with monoclinic structure. The valance band (VB) and conduction band (CB) positions at CuO surface have been measured to be +0.7 and −1.03 eV, respectively, using X-ray photoelectron spectroscopy (XPS) and were found very promising for efficient catalytic properties. The phase purity has been confirmed by Raman spectroscopy and XRD Pattern. The strong light absorption ability in the visible region and an enhanced band gap energy have further confirmed that the synthesized material (CuO nanopetal) is in the nanoscale range. The CuO nanopetals in the presence of $H_{2}O_{2}\;$ has enhanced the degradation efficiency, and maximum degradation efficiency of MB is > 99% within 90 min, which exhibit an efficient catalytic response of the CuO nanopetals. The enhanced catalytic activity is due to a well-defined large surface area of stable CuO nanopetals, efficient charge separation due to the co-operative role of $H_{2}O_{2}\;$ and band structure. These CuO nanopetals have the potential to be used as a room temperature catalyst without using any specific light source. Furthermore, future intentions could be targeted for the development of novel hybrid CuO based nanocomposites with even larger surface areas and to take full advantage of varied surface reactions. This will achieve more interesting properties and promising applications in pollution trace detection and environmental improvement.

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
