# Peer review of "Surface Study of CuO Nanopetals by Advanced Nanocharacterization Techniques with Enhanced Optical and Catalytic Properties"

_nanomaterials, 2020, doi:10.3390/nano10071298_

Round 1

Reviewer 1 Report

Khan et al. report the hydrothermal synthesis and subsequent characterization of CuO nano-petals, which are used for catalytic degradation of methylene blue in aqueous solution.

I consider the manuscript acceptable for publication, after a few minor corrections have been implemented

General:

  • Try to use short sentences. A few sentences go over multiple lines, making it difficult to follow for the reader
  • Between all figures and unit symbols (even %) a blank is required
  • A proof-reading by a native speaker to improve the language-mistakes (tense, plural/singular, phrasing, terminology, …) as this would enhance the readability of the manuscript. Especially misleading wording, as well as the use of a single tense during the whole manuscript should be in the focus
  • All graphical material should be provided in a high resolution. Resolution of many images and schemes in the manuscript is currently not sufficient
  • Please use the same formatting through the complete manuscript. In various paragraphs different formats are used
  • Please use FE-SEM through the whole manuscript, as well as UV-VIS
  • Please define the abbreviation MB at the beginning
  • Line 29-30: This statement needs further explanation, especially in the manuscript itself and not only in the abstract
  • Line 31: The authors state, their nano-petals could be suitable for renewable energy. So far, this is just use of a buzz-word, as this concept is not detailed further in the manuscript
  • Line 71: Please define or rephrase “…nature of the activities of ?2?2”
  • The authors should introduce the reader at the beginning of the introduction better to the reason and motivation for this study
  • Line 101: Please remove “Materials”
  • Line 114: Room temperature according to IUPAC is 298.15 K. However, there is no “normal” room temperature. If the used temperature was different from 298.15 K, please give the exact value
  • Line 118: The phrase “drying in a nitrogen stream for 2 hours at 60°C in a vacuum oven” is not clear – was there a flow of N2 through a vacuum oven, or did the drying take place under reduced pressure, etc.
  • For figure 2 a comparison with the P-XRD pattern of the database would be highly desirable
  • Line 202: Copper should be (Cu)
  • Line 208: 3d9 instead of subscript
  • Line 418: Please give an explanation for the statement, the bandgap between 0.7 and -1.3 eV would be very promising for efficient catalytic properties
  • Please give a short outlook at the end of the conclusion, stating possible next steps towards applicability, and if the degradation would probably also work for other pigments

Author Response

Response to Reviewer # 1:

[Nanomaterials] Manuscript ID: nanomaterials-780526

Title:  “Surface Study of CuO Nanopetals by Advanced Nanocharacterization Techniques with Enhanced Optical and Catalytic Properties”

Authors Reply:

First of all, we would like to thank the editorial board and all the respectable reviewers of the “Nanomaterials” journal for their valuable comments and suggestions. In light of these useful comments, we have tried to improve our manuscript. We have made corrections and revised the manuscript. The detailed responds to reviewer comments and suggestion is attached

Reviewer 2 Report

The paper introduces very interesting structures of CuO and demonstrates its applications as catalysts for MB degradation. While the exact mechanism to form these kinds of structures is not discussed in the paper, their performance and morphology worth publishing. Some specific comments may include 

•   Other work on CuO supercapacitors and batteries should also be cited, such as DOI: 10.1088/1361-6528/aae5c6, DOI: 10.1002/adma.201705830

.   Line 161: The authors claimed “large surface area and well-defined morphology “ SEM cannot be used to determine the specific surface area.

.   Line 153-175: I don't know what exactly the authors mean by “monodispersed “ when they described powder, but I would recommend commenting on the lack of aggregation between the powder, which is helpful for the target applications

.   Line 190-225: The authors should make it clearer in the text that there is no residual nitrogen from the starting materials. What is the source of carbon in the sample? It is better to discuss that in more details as carbon  my influence the catalytic properties

.   Line 226-245: Please use more references to support the Raman peaks positions

.   Figure 5 b, c , and d: why the core of the particle is darker the borders is it because the nanosheets are folded? Please make it clearer

.   Lines 313-324: it is worth here to compare the measured band gap with similar structures and  materials in the literature

Author Response

Response to Reviewer # 2:

[Nanomaterials] Manuscript ID: nanomaterials-780526

Title:  “Surface Study of CuO Nanopetals by Advanced Nanocharacterization Techniques with Enhanced Optical and Catalytic Properties”

Authors Reply:

First of all, we would like to thank the editorial board and all the respectable reviewers of the “Nanomaterials” journal for their valuable comments and suggestions. In light of these useful comments, we have tried to improve our manuscript. We have made corrections and revised the manuscript. The detailed responds to reviewer comments and suggestion is attached.
